# Psychometric Testing of the Spanish Modified Version of the Mini-Suffering State Examination

**DOI:** 10.3390/ijerph18157821

**Published:** 2021-07-23

**Authors:** Daniel Gutiérrez-Sánchez, Rafael Gómez-García, Isabel María López-Medina, Antonio I. Cuesta-Vargas

**Affiliations:** 1Department of Nursing, Faculty of Health Sciences, University of Málaga, 29071 Málaga, Spain; 2Biomedical Research Institute of Málaga (IBIMA), 29071 Málaga, Spain; rafaelgomez@cudeca.org (R.G.-G.); acuesta@uma.es (A.I.C.-V.); 3Fundación Cudeca, 29631 Málaga, Spain; 4Research Group Nursing and Innovation in Healthcare (CuiDsalud), Department of Nursing, Faculty of Health Sciences, University of Jaén, 23071 Jaén, Spain; imlopez@ujaen.es; 5Department of Physiotherapy, University of Málaga, 29071 Málaga, Spain; 6Institute of Health & Biomedical Innovation, Queensland University of Technology, Brisbane, QLD 4059, Australia

**Keywords:** suffering, palliative care, nursing, validation studies, psychometric properties, advanced cancer

## Abstract

Background: The mini-suffering state examination is a valid and reliable measure that have been used to assess suffering in patients with advanced cancer. The aim of this study was to carry out a psychometric analysis of the Spanish version of the mini-suffering state examination. Method: A validation study was conducted. Seventy-two informal caregivers of deceased patients in palliative care were included in this study. A psychometric testing of content validity, internal consistency, and convergent validity with the Spanish version of the quality of dying and death questionnaire was performed. Results: The original instrument was modified to be used by informal caregivers. The content validity was acceptable (0.96), and the internal consistency was moderate (α = 0.67). Convergent validity was demonstrated (r = −0.64). Conclusion: The Spanish modified version of the MSSE showed satisfactory measurement properties. The Spanish modified version of MSSE can be useful to facilitate screening, monitor progress, and guide treatment decisions in end-of-life cancer patients.

## 1. Introduction

Cancer is a problem whose incidence and prevalence has increased in recent years, and it is the second leading cause of death worldwide [1]. In this sense, palliative care (PC) services can have a positive effect on advanced cancer patients and caregivers [2,3,4]. PC in advanced cancer patients can be relevant for improving patient and caregiver outcomes, including the prevention and relief of suffering [5,6,7]. In this regard, the clinicians play a crucial role in the assessment and relief of suffering of patients and families at the end of life [8]

Suffering is subjective in nature, and it affects the whole person [5]. In this regard, the evaluation of suffering is, by definition, subjective, and is influenced by several factors, such as psychological, social, physical, and spiritual [9,10]. Suffering is difficult to evaluate and define and is one of the most feared symptoms for patients with advanced cancer, and the evaluation of this construct is therefore crucial [10]. In this context, we need measures that allow us to assess this construct so as to gather more evidence [11]. Information from proxy raters can be useful and reliable when the proxy is a family caregiver, and the patient’s consciousness is disturbed [12,13]. Assessing suffering from the perspective of family caregivers of advanced cancer patients can be useful for clinicians and researches.

There is limited information available on assessing suffering in the Spanish culture, and Spanish is one of the most widely spoken languages in the world. To the best of our knowledge, there are no specific instruments to evaluate suffering in Spanish cancer patients from the perspective of family caregivers, which is crucial for its prevention, early management, and treatment [6,11]. The mini-suffering state examination (MSSE) is a valid and reliable measure that has been used to assess suffering in patients with advanced cancer [14]. This questionnaire was cross-culturally adapted into Spanish, but no psychometric testing has been performed [15]. The MSSE is a measure that has been developed to assess suffering in patients with end-stage dementia. While PC has traditionally been an approach globally associated with people with cancer, there is support for expanding PC early in the course of chronic conditions, including for people with dementia [16]. In this context, patients with end-stage dementia are considered patients with PC needs [17]. On the other hand, although this measure was developed to assess suffering in patients with end-stage dementia, it has been used to assess this construct in end-of-life cancer patients in a hospice setting, and evidence for validity and reliability has been demonstrated in this population [14]. In this regard, the aim of this study was to perform a psychometric analysis of content validity, internal consistency, and convergent validity of an instrument for assessing suffering in Spanish population, the MSSE with the Spanish version of the quality of dying and death questionnaire (QODD-ESP-26), a previously validated measure that has been used to assess the quality of dying and death in patients with cancer.

## 2. Materials and Methods

### 2.1. Study Design

A cross-sectional, validation design was used to conduct this study. This study was conducted in two principal phases: (1) instrument modification and (2) psychometric analysis.

#### 2.1.1. Phase 1. Instrument Modification

The instrument modification was carried out in the first phase. The MSSE is an instrument originally used by the staff. Because we have used the instrument for informal caregivers of deceased, advanced-cancer patients, the MSSE needs to be modified prior to being used by this population. An expert panel review was conducted to adapt the instrument and to evaluate the content validity of the MSSE according to the modified Delphi technique [18]. Fourteen panelists (clinicians and researchers experts in PC and informal caregivers of advanced-cancer patients) were invited to participate, and in the end, 8 panelists (3 nurses, 2 physicians, 1 social worker, 1 psychologist, and 1 informal caregiver) participated in the expert panel. The content validity box of the Consensus-based Standards for the Selection of Health Measurement Instruments (COSMIN) was used to assess the content validity of the MSSE [19]. A pilot study was carried out to ensure that the modified version of the MSSE was comprehensible and acceptable.

#### 2.1.2. Phase 2. Psychometric Analysis

The psychometric analysis of MSSE was carried out in the second phase. A psychometric analysis of internal consistency and convergent validity of the MSSE with the QODD-ESP-26 was performed.

### 2.2. Measures

Two instruments were used in this study: the MSSE and the QODD-ESP-26.

#### 2.2.1. Mini-Suffering State Examination

The MSSE is a measure that has been developed to assess suffering in patients with advanced dementia [20]. This measure was based on the Cassell’s concept of suffering, where suffering is a specific state of severe distress associated with events that threaten the intactness of the person, and it can be influenced not only by physical but also by psychological, social, and spiritual factors [5]. Inter-observer reliability and concurrent validity of this questionnaire have been evaluated in patients with dementia [20]. In this context, reliability was satisfactory (Cronbach α values 0.735 and 0.718 for the two physicians) [20]. The κ agreement coefficient was 0.791 [20]. Concurrent validity with the comfort assessment in those dying with dementia (CAD-EOLD) was demonstrated (r = −0.796, *p* < 0.001) [20]. Although this measure was developed to assess suffering in patients with dementia, it has been used to assess this construct in patients with advanced cancer [14]. The internal consistency of MSSE has been evaluated in advanced-cancer patients, as the Cronbach alpha’s value was satisfactory (Cronbach α = 0.738) [14]. This questionnaire was cross-culturally adapted into Spanish, but no psychometric testing has been performed [15]. This measure comprises 10 items that can be rated from 0 (no) to 1 (yes) [15]. These 10 items assess the presence of calmness, screaming, pain, pressure ulcers, malnutrition, eating disorders, performance of invasive procedures, stability of general medical condition, and patient’s suffering according to medical and family opinion. The overall score ranges from 0 to 10 [15]. The overall score can be grouped into three categories: ‘‘low level of suffering’’ (0 to 3), ‘‘intermediate level of suffering’’ (4 to 6), and ‘‘high level of suffering’’ (7 to 10) [15,20].

#### 2.2.2. Spanish Version of the Quality of Dying and Death Questionnaire (QODD-ESP-26)

The QODD-ESP-26 is a valid and reliable instrument to assess the quality of dying and death in the Spanish population [21,22]. Psychometric testing of this measure has been performed with the family caregivers of advanced-cancer patients who have died. In this context, satisfactory values of content validity (content validity index = 0.96), internal consistency (α = 0.88), divergent validity (r = −0.64), and convergent validity (r = −0.61) were obtained [21]. This tool comprises 26 items that are posed to the informal caregivers of the advancer cancer patients who have died [21]. The instrument concerns the quality of dying and death in the last seven days of the patient’s life unless the patient was unresponsive throughout the last seven days, in which case the period rated is the last month of life [21]. The items of this instrument consist of two parts. In the first part, the informal caregiver assesses the frequency (0 = none to 5 = always) or existence (yes or no) of the specific attribute for the patient, and in the second part, the informal caregiver assesses this attribute of the patient’s dying experience on a scale of 0 (‘‘a terrible experience’’) to 10 (‘‘an almost perfect experience’’) [21]. An overall score can be obtained by adding the ratings of the patient’s dying experience, then dividing by the number of items answered, which is divided by 10 and multiplied by 100 [21,23,24]. The overall score ranges from 0 to 100, with higher scores indicating better quality of dying and death [21,25].

### 2.3. Setting, Sample, and Procedure

Participants were recruited from two PC centers in Málaga, Spain (Cudeca Foundation and Regional University Hospital of Málaga). Data on the informal caregivers and advanced-cancer patients who died were obtained between January and October 2016. The inclusion criteria were: (1) Spanish-speaking, adult, informal caregivers who had cared for an adult patient; (2) those who had signed an informed consent; and (3) informal caregivers of deceased patients included in the PC program. The exclusion criteria used was cognitive impairment.

Potential informal caregivers who met the inclusion criteria were included. One to six months after the death of a patient, a letter of condolence was sent to the family caregiver of the deceased. After that, informal caregivers were contacted to inquire about their availability to participate in the present research. Informal caregivers who wanted to participate in this study were informed about the study’s method and procedure as well as the protection of their personal data. After signing an informed consent, the informal caregivers received the documents to fill out. All documents were returned to the PC center once they were filled out.

### 2.4. Data Analysis

A descriptive analysis was performed to estimate the sociodemographic variables. We determined the distribution and normality of the sample by performing a one-sample Kolmogorov–Smirnov (KS) test. A statistical psychometric analysis was carried out to estimate the content validity, internal consistency, factor structure, and criterion validity of the MSSE. The Lawshe method was used to obtain a content validity index [26,27]. Cronbach’s α coefficients and intraclass correlation coefficient type 2.1 (ICC_2.1_) were calculated to obtain the internal consistency of the MSSE [28].

Criterion validity was determined through the use of the MSSE and QODD-ESP-26 total score. The Pearson correlation coefficient was used. The data analysis was carried out using SPSS version 24.

### 2.5. Ethical Aspects

This study is an integral part of a larger project. The project was approved by the Ethics Committee of the Area of Málaga and the Ethics Committee of Area of Costa del Sol of Málaga (Spain) in January 2016 (Project identification code: 001_ENE_PI_2_QODD-15). Each participant received a detailed explanation of the study and gave written informed consent before participation. Confidentiality was assured by separating clinical data from personal identification data.

## 3. Results

### 3.1. Sample

The KS test (*p* = 0.743) indicated that the sample was normally distributed. One hundred and seventy-six informal caregivers were identified. Eighty-five questionnaires were returned, and thirteen cases were eliminated due to the high percentage of unanswered items (more than 25%). A total of 72 informal caregivers of deceased patients were included in this study. The majority were females (52) and had a mean age of 51.11 years old (±11.69) (Table 1). The deceased, advanced-cancer patients were more frequently females (39) and had a mean age of 72.11 years old (±12.46) (Table 1). The mean MSSE overall score was 4.13 (±2.16).

### 3.2. Phase 1. Instrument Modification

Panelists indicated that some MSSE items needed to be modified to be used by informal caregivers. Four items—“pressure ulcers”; “eating disorders”; “performance of invasive procedures”; and “unstable medical condition”—were modified to clarify their meanings as follows: “decubitus ulcers/pressure ulcers”; “eating disorders (e.g., difficulty to swallow, anorexia) ”; “performance of invasive procedures (e.g., analytics, transfusions, catheters, hemodialysis, mechanic ventilation)”; and “unstable medical condition (deterioration of health condition due to progress of the terminal illness, infections, etc.)”. A period of time concerning the suffering in the last week of the patient’s life was added as follows: “Assess the presence of the following elements during the last 7 days of the patient’s life”. A content validity index (CVI) was obtained. The CVI was 0.96.

A pilot study was conducted with 32 informal caregivers of deceased patients to ensure that the modified version of the MSSE was comprehensible and acceptable. The final, modified Spanish version of the MSSE for informal caregivers was demonstrated to be understandable and acceptable.

### 3.3. Phase 2. Psychometric Analysis

#### 3.3.1. Internal Consistency

The overall internal consistency of the MSSE was 0.67, and the ICC was 0.67 (95% CI 0.53 to 0.77) (Table 2).

#### 3.3.2. Convergent Validity

Convergent validity was determined from the relationship between the MSSE and QODD-ESP-26. A fair and inverse correlation of r = −0.64 (*p* < 0.001) was obtained.

## 4. Discussion

The relief of suffering is traditionally considered one of the main goals at the end of life [29]. In this context, PC services plays a crucial role in relieving the suffering of advanced-cancer patients and families [2,3].

Suffering at the end of life is difficult to evaluate and define [10]. In this context, evaluations given by informal caregivers who cared for the advanced-cancer patients during the last days of their lives have been used as indirect measurements (proxy) to evaluate the patient’s suffering [19].

Validated instruments for assessing suffering can be useful to facilitate screening, monitor progress, and guide treatment decisions in end-of-life cancer patients [30]. In this regard, evidence for the validity of the modified Spanish version of MSSE has been provided in the current study.

This study is the first psychometric analysis of the modified Spanish version of MSSE, and Spanish is one of the most commonly spoken languages in the world. The psychometric analysis of this questionnaire was performed satisfactorily. In this context, the modified Spanish version of MSSE showed adequate values for content validity, internal consistency, and convergent validity [26,27,31,32].

This instrument has been modified, after consensus, to be used with the informal caregivers of the deceased, advanced-cancer patients, and it has been demonstrated to be understandable and acceptable.

The modified Spanish version of MSSE showed adequate values of content validity (CVI = 0.96), which indicates that all items are relevant for the suffering measurement [26,27].

The internal consistency of the MSSE was acceptable (α = 0.67), which indicates a moderate homogeneity of the items [30]. These values are lower than those reported in others studies (Cronbach’s α = 0.738) [14].

Convergent validity with a previously validated instrument for assessing quality of dying and death was analysed. In this regard, a fair and inverse correlation of r = −0.64 (*p* < 0.001) was obtained for this psychometric property.

The values of suffering assessed with MSSE were satisfactory 4.13 (±2.16). In comparison with other studies in a PC population, our results indicated a lower level of suffering (4.57 ± 2.57) [14].

The assessment of suffering of cancer patients at the end of life will allow us to improve care in this population, as the measurement of this construct is crucial. Thus, it is necessary to have reliable and valid instruments for assessing suffering in the advanced-cancer population. In this context, the present study contributes to outcome evaluation of cancer patients at the end of their lives.

### 4.1. Study Strengths and Limitations

This is the first study in which the measurement properties of the modified Spanish version of MSSE have been analysed. The psychometric analysis of this questionnaire was performed satisfactorily.

There are some limitations in this study. Psychometric properties, such as structural validity, inter-rater reliability, and sensitivity to change, have not been analyzed in this study. Moreover, evaluations given by informal caregivers have been used to evaluate the patient’s suffering. In this context, the patient’s suffering assessment could be affected by the time from death to the assessment, with the optimal timing to gather data from family caregivers being essential. In this regard, an interval of one to six months after the patient’s death was used in this study, a similar period of time to that used in other proxy reports after death [22]. Further studies should include an analysis of other measurement properties in different Spanish clinical populations.

### 4.2. Implications for Future Research

The current study demonstrates that the modified Spanish version of MSSE is a valid and reliable measure to evaluate suffering in advanced-cancer patients. This instrument can be useful to facilitate screening, monitor progress, and guide treatment decisions in PC services. The evaluations carried out by family members can be used as an indirect measure of the suffering, allowing clinicians and researchers to evaluate this construct. Suffering research can increase our understanding of end of life, which is essential to achieving a “good death” in advanced-cancer patients. Further longitudinal studies that evaluate interventions for relieving suffering in the advanced-cancer population are needed.

## 5. Conclusions

The prevention and relief of suffering are crucial to the well-being of patients with advanced cancer and their families. The assessment of suffering of cancer patients at the end of life will allow us to improve care in this population, and validated instruments for assessing suffering can be useful to facilitate screening, monitor progress, and guide treatment decisions in end-of-life cancer patients. In this regard, we need instruments that allow us to measure suffering so as to gather more evidence. This study is the first psychometric analysis of the modified Spanish version of the MSSE. Evidence for the validity of the modified Spanish version of the MSSE has been provided in the current study. Thus, this is a valid and reliable measure for assessing suffering in the advanced-cancer population. Further studies should include an analysis of other measurement properties in different Spanish clinical populations.

## Figures and Tables

**Table 1 ijerph-18-07821-t001:** Demographic data of family caregivers and patients (*n* = 72).

Characteristics	Frequency
Caregivers	
Age (mean, SD)	51.11 (±11.69)
Gender	
Male	20
Female	52
Relationship to patient	
Spouse	11
Son	11
Daughter	29
Sister	4
Other relative	17
Number of days between death and interview (mean, SD)	172 (±55)
Patients	
Age (mean, SD)	72.11 (±12.46)
Gender	
Male	33
Female	39

Note: SD, standard deviation.

**Table 2 ijerph-18-07821-t002:** Descriptive statistics and internal consistency for items from the MSSE.

MSSE Items	Corrected Item—Total Correlation	Cronbach’s Alpha If Item Deleted
Not calm	0.33	0.64
Screams	0.33	0.64
Pain	0.25	0.66
Decubitus ulcers	0.20	0.67
Invasive actions	0.25	0.66
Unstable medical condition	0.31	0.65
Suffering according to medical opinion	0.64	0.56
Suffering according to family opinion	0.57	0.57

Note: MSS, mini-suffering state examination; SD, standard deviation.

## Data Availability

The data presented in this study are available on request from the corresponding author.

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
