# Peer review of "Psychometric Testing of the Spanish Modified Version of the Mini-Suffering State Examination"

_ijerph, 2021, doi:10.3390/ijerph18157821_

Round 1

Reviewer 1 Report

A well structured manuscript but authors should update the reference list. Refrences should be of the last 5-7 years.

Author Response

  • Authors: Thank you. We have removed old references and provided updated references where appropiate.

Reviewer 2 Report

The article reports the translation and psychometric testing of the Spanish modified of MSSE. Some issues regarding the methodology of the study need clarification/modification:

    1. It is unclear why the scale originally developed for caregivers of dementia can be modified for caregivers of palliative care.
    2. While the psychometric properties of the MSSE in other samples were provided, the details of the MSSE including the conceptual definition of the construct is measured by the scale, number of items, any subscales within the measure, the response alternatives, should be provided.
    3. Sample size (n=72) is too small for exploratory factor analysis and confirmatory factor analysis, which might lead to misleading results. Please provide justification for the sample size. Indeed, the KMO of 0.607 only indicates mediocre and only 18.394% of the variation was explained by the one-factor solution in the EFA clearly indicates a poor fit to the data. In addition, the selection of one-factor solution is questionable – the first 4 eignevalues were greater than 1 while the criterion of factor retention is eignevales >=1. Also, binary items are included in the MSSE, maximum likelihood estimation method in EFA is inappropriate as normality of the data at item level was not met. Estimation method for binary variables should be used instead. A sample size of 72 is clearly not enough for obtaining accurate estimates for CFA.  
    4. Divergent validity has to show the two constructs measured by the scales were different, that is with little or no relationship. So, a large negative correlation between MSSE and QODD-ESP-26 (r = -0.64) indicates the two variables are moderately correlated.
    5. A KS test was performed but it is unclear the KS test was performed on which variable? ‘Asymp Stat (2-tailed)’ should be replaced by ‘p-value’.

Reviewer 3 Report

Accept in present forma.

Author Response

Reviewer: 3

Comments to the Author: Accept in present forma.

  • Authors: Thank you.

Round 2

Reviewer 2 Report

Although the authors have attempted to address the concerns but unsatisfactorily as many questions remained unanswered. The authors are strongly advised to consult an expert in psychometric testing.

  1. The appropriateness of applying the MSSE in the caregivers of palliative care patients is provided in the response letter, not in the text of the manuscript.  Consequently, readers who have similar concern could not have a chance to see the rationale.
  2. EFA- the authors did not explain why PCA was used. Although many researchers have argued that the use of PCA and MLE in EFA will produce similar results but other researchers had argued PCA uses a different conceptual framework of formative model which assumes the latent composite is caused by the indicators. In addition, the authors' response to sample size determination is not satisfactory. It is important to ensure the sample size is large enough to ensure the accuracy of the parameter estimates in quantitative study. One way to overcome is to spend more time in subject recruitment before reporting the premature results. Please show the sample size is large enough for the current analysis. The question about the use of EFA on binary items was unanswered.
  3. The main issue in EFA is (i) the low factor loadings of some of the items, and (ii) the low percentage of variances explained. The authors should NOT use only one criterion to determine the number of factors in EFA. We usually use several criteria including eigenvalues, scree plot, size of factor loadings and % of variance explained. Factor loadings of Item5 on malnutrition and Item 6 on eating disorder are very small, indicating they do not belong to this factor. The % of variance explained also indicates that the one-factor model does not provide a good representation to the data - more factors are needed to explain the variations among the items. 
  4. The correlational analysis of MSSE with QODD-ESP-26 is attempted to look for convergent validity, not criterion nor divergent validity. Please check the definition of the different type of validities.
  5.  Limitations regarding the EFA were not mentioned, including small sample size and the use of binary variable in EFA.

Author Response

Comments and Suggestions for Authors

Although the authors have attempted to address the concerns but unsatisfactorily as many questions remained unanswered. The authors are strongly advised to consult an expert in psychometric testing.

  1. The appropriateness of applying the MSSE in the caregivers of palliative care patients is provided in the response letter, not in the text of the manuscript. Consequently, readers who have similar concern could not have a chance to see the rationale.
  • Authors: Thank you. According to your comment, we have added this information in Introduction section, and two new references (16 and 17).

  1. EFA- the authors did not explain why PCA was used. Although many researchers have argued that the use of PCA and MLE in EFA will produce similar results but other researchers had argued PCA uses a different conceptual framework of formative model which assumes the latent composite is caused by the indicators. In addition, the authors' response to sample size determination is not satisfactory. It is important to ensure the sample size is large enough to ensure the accuracy of the parameter estimates in quantitative study. One way to overcome is to spend more time in subject recruitment before reporting the premature results. Please show the sample size is large enough for the current analysis. The question about the use of EFA on binary items was unanswered.

  • Authors: We appreciate your comment. Although EFA would have added further value to the present manuscript, KMO showed that sample size not wide enough for EFA. In this context, we have removed the EFA from the paper and we have added a new statement to limitations section.

  1. The main issue in EFA is (i) the low factor loadings of some of the items, and (ii) the low percentage of variances explained. The authors should NOT use only one criterion to determine the number of factors in EFA. We usually use several criteria including eigenvalues, scree plot, size of factor loadings and % of variance explained. Factor loadings of Item5 on malnutrition and Item 6 on eating disorder are very small, indicating they do not belong to this factor. The % of variance explained also indicates that the one-factor model does not provide a good representation to the data - more factors are needed to explain the variations among the items.

  • Authors: We appreciate your comment. As we commented before, although EFA would have added further value to the present manuscript, KMO showed that sample size not wide enough for EFA. In this context, we have removed the EFA from the paper and we have added a new statement to limitations section.

  1. The correlational analysis of MSSE with QODD-ESP-26 is attempted to look for convergent validity, not criterion nor divergent validity. Please check the definition of the different type of validities.
  • Authors: Thank you. According to your comment, we have used the term “convergent validity” instead of “criterion validity” throughout the paper.

  1. Limitations regarding the EFA were not mentioned, including small sample size and the use of binary variable in EFA

  • Authors: We thank you for the comment. We have removed EFA from the paper.